# Characterizing Muscle Tissue Quality Post-Stroke: Echovariation as a Clinical Indicator

**DOI:** 10.3390/jcm13247800

**Published:** 2024-12-20

**Authors:** Borhan Asadi, Clara Pujol-Fuentes, Alberto Carcasona-Otal, Sandra Calvo, Pablo Herrero, Diego Lapuente-Hernández

**Affiliations:** 1Department of Physiatry and Nursing, Faculty of Health Sciences, University of Zaragoza, 50009 Zaragoza, Spain; basadi@iisaragon.es (B.A.); cpujol@iisaragon.es (C.P.-F.); acarcasona@unizar.es (A.C.-O.); sandracalvo@unizar.es (S.C.); d.lapuente@unizar.es (D.L.-H.); 2iHealthy Research Group, Instituto de Investigación Sanitaria (IIS) Aragon, University of Zaragoza, 50009 Zaragoza, Spain

**Keywords:** stroke, ultrasonography, echotexture analysis, muscle tissue, Heckmatt scale, gastrocnemius medialis

## Abstract

**Background/Objectives:** Strokes remain a major global health concern, contributing significantly to disability and healthcare costs. Currently, there are no established indicators to accurately assess the degree of muscle tissue impairment in stroke-affected individuals. However, ultrasound imaging with an echotexture analysis shows potential as a quantitative tool to assess muscle tissue quality. This study aimed to identify specific echotexture features in the gastrocnemius medialis that effectively characterize muscle impairment in post-stroke individuals. **Methods**: An observational study was conducted with 22 post-stroke individuals. A total of 21 echotexture features were extracted and analyzed, including first-order metrics, a grey-level co-occurrence matrix, and a grey-level run length matrix. The modified Heckmatt scale was also applied to correlate with the most informative echotexture features. **Results**: Among the features analyzed, echovariation (EV), echointensity, and kurtosis emerged as the most informative indicators of muscle tissue quality. The EV was highlighted as the primary feature due to its strong and significant correlation with the modified Heckmatt scale (r = −0.81, *p* < 0.001) and its clinical and technical robustness. Lower EV values were associated with poorer muscle tissue quality, while higher values indicated better quality. **Conclusions**: The EV may be used as a quantitative indicator for characterizing the gastrocnemius medialis muscle tissue quality in post-stroke individuals, offering a more nuanced assessment than traditional qualitative scales. Future studies should investigate the correlation between the EV and other clinical outcomes and explore its potential to monitor the treatment efficacy, enhancing its applicability in clinical practice.

## 1. Introduction

Strokes are a significant health issue with a profound socioeconomic impact, being one of the leading causes of disability worldwide [1]. Recent data indicate that approximately 13.7 million people experience a stroke annually, with substantial costs associated with long-term care and rehabilitation [2]. The effective management of strokes implies objective, accurate, and reproducible measurement tools that can be integrated into clinical practice to assess patient progress and optimize therapeutic outcomes to ensure that interventions are appropriately tailored to individual patient needs, enhancing their recovery and quality of life.

Over the years, a texture analysis has emerged as a key technique to characterize muscle composition in various diagnostic imaging technologies. A texture image is defined by the mathematical evaluation of pixel intensities and their spatial distribution within a region of interest (ROI) [3,4]. Statistical methods, classified into first-order (individual pixel intensities), second-order (ratios between pixel pairs), and higher-order features, provide information on tissue patterns [5,6]. However, reproducibility and validation problems have limited the clinical application of these methods [7,8,9,10].

In the concrete case of ultrasound (US), it has emerged as a readily accessible, non-invasive, and cost-effective method for assessing skeletal muscle composition [11]. Furthermore, the acoustic power levels employed in US devices are designed to minimize the risk of adverse biological effects [6]. To date, US image analyses of muscle tissue have mainly focused on muscle architecture characteristics, including the cross-sectional area, thickness, fascicle length, and fascicle pennation angle [12,13]. Nonetheless, muscle echotexture features have also been explored in the context of certain medical conditions, such as amyotrophic lateral sclerosis [14], myofascial pain syndrome [15], or low back pain [16], as well as in healthy tissue [6]. However, considerable research is still required before these features can be reliably implemented in clinical practice.

Regarding muscle echotexture features in individuals who have suffered a stroke, the most extensively studied feature is the echogenicity/echointensity (EI), defined as the mean pixel brightness within the muscle fascia boundaries [17]. This feature appears to be altered in this population due to changes in histopathological features like atrophy and the fibro-adipose replacement of hypoechoic contractile elements [18], resulting in a higher EI due to the increased reflection of sound waves [19,20,21]. The EI can be assessed both qualitatively and quantitatively. The original or modified Heckmatt scale can be used for a qualitative analysis, which simply classifies the image into four possible degrees of impairment without an extensive analysis [22]. This scale has demonstrated a good reliability and validity in evaluating muscle changes in patients with spasticity [23]. Several free software programs for quantitative analyses exist to calculate the EI, but their clinical application is still limited due to the complexity and variability of the measurements. This variability is influenced by factors such as the type of US device, the settings, the operator technique, and image artifacts [24,25,26].

Another promising parameter for characterizing muscle composition in neuromuscular or neurological disorders is echovariation (EV), as a measure of the EI variation [27,28]. However, its application in the context of strokes has not yet been investigated. Additionally, a critical issue in existing studies on other conditions is the substantial variability of the echotexture features employed and the resulting data, often leading to the occurrence of spurious findings [29]. Consequently, this study aimed to identify the most informative echotexture features for characterizing gastrocnemius medialis muscle tissue in stroke-affected individuals. This effort could assist future research in selecting these features and evaluating their potential as reliable indicators for assessing the extent of muscle impairment and determining any relationships with other clinical outcomes in stroke-affected individuals.

## 2. Materials and Methods

### 2.1. Study Design

This observational study included stroke-affected individuals and analyzed the echotexture features of the gastrocnemius medialis muscle. The study followed the STROBE recommendations for observational studies [30] and the Declaration of Helsinki [31].

The Aragon Ethics Committee (PI24/030) approved the study and registered on ClinicalTrials.gov (NTC06411587, https://clinicaltrials.gov/study/NCT06411587, accessed on 7 February 2024). This study was based on a subset of data from the trial registered under the specified ClinicalTrials number, in which an analysis of the echotexture from various US images was conducted.

### 2.2. Participants

Stroke-affected individuals were recruited at the Hospital Clínico Universitario Lozano Blesa from February to April 2024 by a physician specializing in physical medicine and rehabilitation. During consultations, the specialists assessed the patients based on the predefined inclusion and exclusion criteria. The inclusion criteria were as follows: (1) to be aged over 18 years; (2) to have a stroke diagnosis confirmed by computed tomography (CT) or magnetic resonance imaging (MRI); (3) to have more than 6 months of evolution since the onset of the stroke; and (4) to be able to walk independently, with or without assistance. The exclusion criteria were the following: (1) to have other neurological disorders (e.g., ataxia or dystonia); (2) to have undergone surgical intervention in the lower limb; and (3) to have other medical conditions that may interfere with data collection.

Patients meeting the criteria were given detailed explanations about the study, including its purpose, potential benefits, and associated risks. Those interested in participating were given an informed consent form, and any questions or concerns were addressed. Upon obtaining written informed consent, the patients were scheduled for an appointment to be evaluated at the Hospital Clínico Universitario Lozano Blesa.

### 2.3. Data Collection

US images of the gastrocnemius medialis muscle were obtained using the Butterfly iQ+ portable US system, specifically designed for external US imaging, manufactured by Butterfly Network, Inc. The imaging protocol utilized a standardized preset of a 50% gain and a 5 to 7 cm depth. The scans were conducted by a physiotherapist experienced in using the US in stroke-affected individuals. The images were taken with the participant seated, with 90° of knee flexion. The probe was placed at a point located proximally at 30% of the distance between the medial condyle of the tibia and the medial malleolus [32], while adjusting the angle from horizontal to obtain the best possible image (usually around 35–45° from horizontal). Six images were taken in total, three for each lower limb of the participant.

### 2.4. Variables and Data Analysis

The whole process, from data collection to analysis, is summarized in Figure 1. After the physiotherapist (C.P.-F) collected the US images, they were uploaded into a cloud server and analyzed by an independent researcher (B.A.). Subsequently, two expert physiotherapists (D.L.-H and P.H.) manually selected the ROI for each image by consensus, including the maximum gastrocnemius medialis muscle tissue, while excluding superficial and deep fascia [33] (Figure 2). Images with an unclear delineation of the fascia or significant artifacts that could affect the image analysis were excluded. This ROI selection process followed a standardized image-acquisition protocol to minimize the variability. A specific ROI size was not predefined to avoid bias. Instead, the ROI was selected manually for each image, aiming to include the maximum amount of muscle tissue. This approach ensured a more representative analysis, as chronic conditions are expected to affect the muscle uniformly, unlike acute injuries, where selecting a specific ROI is necessary to target localized damage.

Three main types of features were extracted from each of the US images (Table 1) without any image pre-processing:Histogram-based (first-order features): the EI [25], EV [28], variance, standard deviation, skewness, and kurtosis [34]. These features were computed directly from the pixel matrix of the images.Grey-level co-occurrence matrix (GLCM): the correlation, dissimilarity, contrast, homogeneity, angular second moment (ASM), energy, maximum probability, entropy [35], cluster shade, and cluster prominence [36]. The GLCM was calculated using the skimage.feature library (https://scikit-image.org/docs/stable/api/skimage.feature.html, accessed on 20 October 2024), with distances = 5, angles = 0, and levels = 256.Grey-level run length matrix (GLRLM): the short run emphasis (SRE), long run emphasis (LRE) [37], grey-level Uuniformity (GLU), run length uniformity (RLU) [38], and run percentage (RPC) [39]. The GLRLM was calculated using a custom library (https://pypi.org/project/glrlm/, accessed on 20 October 2024). The final output of the GLRLM was obtained for all angles: 0°, 45°, 90°, 135°, and 180°, using a level parameter of 16. The features were then extracted from each matrix, and the final output was derived by averaging the features across all angles.

The methodology to select the most informative features for assessing muscle impairment or muscle tissue quality assumed that each feature’s top and bottom 10 values should correspond to the images with the most and the least affected tissue. Each feature’s images were ordered from the highest to the lowest values for this exploratory analysis and graphically represented. The 20 images with the top and bottom 10 values were then selected for each feature, and two physiotherapists (D.L.-H and P.H.) assessed the degree of muscle impairment, considering that more significant impairment was indicated by an increased loss of muscle echotexture (e.g., diffuse areas). These two physiotherapists underwent joint training sessions before the analysis to standardize the evaluation process. Following training, each evaluator independently assessed the images while blinded to the participants’ clinical outcomes to ensure objective evaluations solely based on image-derived features. Features were selected only when the top and bottom 10 values corresponded 100% with high or low muscle impairment and when the images with high or low quality were in the opposite extreme for both evaluators. For example, features characterizing 100% of the images as highly affected for both the top and bottom 10 values were not considered informative features. Following this process, the two evaluators summarized the number of images grouped at each extreme according to the degree of impairment, allowing them to determine the most informative features.

After this, a third independent physiotherapist (C.P.-F) classified the 130 US images based on the four grades of the modified Heckmatt scale, which has been proven to be both reliable and valid for assessing spastic muscle and is easier to use in clinical practice compared to the original Heckmatt scale [23]. The relationship between the most informative selected features and the modified Heckmatt scale was examined with Pearson’s correlation coefficient. The correlations were assessed based on established criteria [40]: (1) correlations greater than 0.80 were considered very strong; (2) correlations between 0.60 and 0.80 were considered moderately strong; (3) correlations between 0.30 and 0.50 were considered moderate; and (4) correlations below 0.30 were considered weak. Before the correlation analysis, all the data were normalized using Min-Max normalization to ensure that the variables were on a comparable scale. The *p*-values for each correlation coefficient were calculated to assess the statistical significance, with a threshold of *p* < 0.05 indicating significance.

An additional secondary analysis was carried out after identifying the most informative features for muscle tissue characterization through the preliminary exploratory process and correlation analysis. This analysis was based on the available scientific evidence and/or the inherent characteristics of each feature to select only one as the reference feature. The distribution of this feature was mathematically represented using a function. This approach facilitated the characterization of the extent of muscle tissue impairment in individuals who have experienced a stroke. Additionally, Pearson’s correlation coefficient between the reference feature and each of the other 20 extracted features was calculated, following the same established criteria as before [40]. Features exhibiting a correlation coefficient greater than 0.8 with the reference feature and a *p*-value < 0.05 were selected as possible features of interest. These features were chosen due to their potential to complement the reference feature, indicating the strongest and statistically significant associations.

Finally, the mean and standard deviation of each of the four grades of the modified Heckmatt scale were calculated using the reference feature values, allowing its dispersion to be visualized along the distribution graph. In addition, to determine whether statistically significant differences existed between the groups, an ANOVA test or a Kruskal–Wallis test was performed, depending on the normality of the data distribution. Normality was assessed by examining histograms and Q-Q plots. If statistically significant differences (*p* < 0.05) between groups were detected, a post hoc analysis was performed using Tukey’s honest significant difference or the Bonferroni correction to identify which specific pairs of groups showed significant differences.

Data pre-processing and visualization, the image analysis, and the machine learning analysis were performed in Python, version 3.11.5, using libraries such as numpy, os, matplotlib, random, string, PIL, skimage.feature, mahotas.features.texture, cv2, pandas, scipy.stats, scipy.optimize, sklearn.metrics, and sklearn. The programming environment used was JupyterLab, version 4.0.11. The statistical analyses were performed in RStudio, version 2024.09.0.

## 3. Results

A total of 130 ROIs extracted from US images of the gastrocnemius medialis muscles of both the lower limbs of 22 stroke-affected individuals were analyzed. The sample consisted of 16 (72.73%) men and 6 (27.27%) women, with a mean age of 64.32 ± 13.25 years. The distribution of time since the stroke onset among the participants was as follows: five participants (6 to <12 months), eleven participants (12 to <24 months), five participants (24 to <36 months), and one participant (36 to <40 months). Of the participants, 5 had experienced a hemorrhagic stroke and 17 had an ischemic stroke. Notably, only one participant was on spasticity medication during the study.

Following the selection of the ROIs and a subsequent analysis of the extracted features, three features (EV, EI, and kurtosis) were identified based on the methodology described. These features were the most informative for assessing gastrocnemius medialis muscle tissue impairment in individuals who had experienced a stroke. Specifically, reduced values of EV and kurtosis indicated greater muscle tissue impairment, while the EI had the highest values (Table 2). After the image classification by the third physiotherapist using the modified Heckmatt scale, the analysis revealed a very strong and significant correlation for the EV (r = −0.81, *p* < 0.001) and the EI (r = 0.87, *p* < 0.001), but not for kurtosis (r = −0.37, *p* < 0.001).

The mathematical function for the distribution of the EV was calculated and fitted using the model *y* = *ae^bx^
*+ *ce^dx^* + *E* with parameter estimates of *a* = 37.13, *b* = −0.14, *c* = 132,052.16, *d* = −2 × 10^−6^, and *E* = −131,990.30. This model demonstrated a high degree of accuracy, as evidenced by an R^2^ error rate of 0.993, indicating that the function effectively represents the distribution of the EV. The graphical representation of this distribution is shown in Figure 3, where lower EV values along the Y-axis correspond to images in which experts identified the muscle tissue as having a poorer quality and more significant impairment (as illustrated in Appendix A). Conversely, higher EV values were associated with images of less impaired muscle tissue, suggesting an improved quality (Appendix A). The location of each of the US images shown as an example of low and high EV along the distribution in Figure 3 is presented as Appendix A.

The correlations between the EV and the other features are detailed in Table 3. Among these features, only the EI showed a Pearson’s correlation coefficient greater than 0.8, demonstrating a statistically significant relationship (*p* < 0.05) with a value of −0.90. This indicates a strong inverse correlation, suggesting that, as the EI increases, the EV decreases, and vice versa. The correlation plot illustrating the relationship between the EI and the EV is presented in Figure 4. In this plot, the data were sorted by the EV in descending order, and the values were subsequently normalized on a scale from 0 to 1 (Figure 4). Additional correlations are provided in the Appendix A).

After assessing the normality of the data, the Kruskal–Wallis test was performed to assess any differences that existed between the EV values according to the four grades of the modified Heckmatt scale, which revealed statistically significant differences between the four grades (*p* < 0.001) (Table 4). A subsequent post hoc analysis, using the Bonferroni correction, confirmed that significant differences existed between all possible pairs of groups, with each comparison reaching statistical significance (*p* < 0.001) (Table 4). Figure 5 illustrates the distribution of EV values across the four grades of the modified Heckmatt scale. For each grade, the mean and standard deviation are shown, providing a clear visualization of the distribution and variability of the data (Figure 5).

## 4. Discussion

One of the existing needs in the analysis of muscle tissue quality is to have an indicator that can be used to quantify the degree of muscle impairment more objectively than qualitative scales such as the original or modified Heckmatt scale, which has been the most widely used to date, despite its limitations. An US is cost-effective in assessing muscle tissue and could serve as an objective tool [41]. In line with our aim to identify the most informative echotexture features for characterizing the degree of muscle tissue impairment in the gastrocnemius medialis of stroke-affected individuals, the findings of our study suggest that the EV, EI, and kurtosis may be the most informative features initially.

Considering the correlation between the modified Heckmatt scale and the three most informative features, the EV and EI seemed the most informative. Different factors were considered to propose only one feature that would be more easily transferable to clinical practice, resulting in the EV being considered the most suitable feature for several reasons: (1) previous studies have consistently identified the EV as a significant metric for assessing muscle tissue across various neuromusculoskeletal conditions; (2) the EV includes the EI in its calculation, as it is defined as the standard deviation of pixel intensity divided by the mean pixel intensity (which constitutes the EI); (3) the EV can be considered a relative measure that is inherently less susceptible to variations in the US image and device settings, unlike the EI, which is significantly influenced by settings such as the gain; (4) the EV is a better measure of the overall relative variability when compared with kurtosis, that takes into account only the extreme values, losing the information between the extremes; and (5) when the EV values were analyzed, there were significant differences between the different grades in the modified Heckmatt scale, which supports the use of this indicator as a complement to quantify the degree of muscle impairment objectively. The decision to retain only one feature was made to enhance its clinical applicability and facilitate its transferability to practical settings.

Based on a mathematical model and considering that the pixel intensity values from greyscale images may range from 0 (black) to 255 (white), when the image is uniform (i.e., standard deviation—σ—is small) and the mean (µ) approaches its maximum value of 255 (all pixels white), the EV is at its lowest. Conversely, when the image is more heterogeneous (i.e., σ is big) and µ approaches its minimum value (all pixels blacker), the EV is at its highest. These observations led us to hypothesize that lower EV values correspond to poorer muscle tissue quality, while higher EV values indicate better muscle tissue quality. Additional details regarding this hypothesis are provided in Appendix A.

This hypothesis recognizes the relationship between EV and muscle tissue quality without attempting to restrict the EV range artificially. Although this study was performed specifically on stroke-affected individuals, we believe its underlying principles can potentially be extended to assessing healthy muscles and muscles in other populations with different pathologies/conditions. However, further research into its use is needed to validate the applicability and determine with confidence the generalizability of this model to other populations, including the healthy population, in an attempt to define a normalized score for muscle tissue.

The proposed model offers a robust framework for quantitatively assessing muscle tissue quality in stroke-affected individuals, addressing limitations inherent to the modified Heckmatt scale, which categorizes impairment into four discrete grades. By providing a more detailed measurement of muscle tissue quality, our model enables physiotherapists to detect subtle variations in muscle tissue that categorical scales may overlook. This capability is crucial for tracking changes over time, responding to therapeutic interventions, and objectively evaluating injury progression and prognoses. For instance, the modified Heckmatt scale might categorize two images as reflecting the same degree of muscle impairment, despite variations in their EV values. Though clinically relevant, such differences remain undetected by the modified Heckmatt scale or even through visual inspection by clinicians. Therefore, using an indicator like the EV may provide additional value in clinical practice compared to a clinical scale like the Heckmatt scale, since it allows for the more granular or objective quantification of muscle impairment. There is still controversy about the modified Heckmatt scale, with recent studies suggesting that it would be better to reduce it from four to three grades [42].

Although the modified Heckmatt scale is based on the EI, we propose the EV as a more reliable indicator than the EI based on our results. Our results indicate that higher EV values are associated with healthier or higher-quality muscle tissue, while lower EV values correspond to more severely impaired muscle tissue in stroke-affected individuals. Conversely, the relationship is inverted for the EI; higher EI values mean a poorer muscle tissue quality, whereas lower EI values indicate a greater similarity to healthy muscle tissue. The strong correlation between the EV and EI allows us to assert that, as muscle tissue becomes increasingly impaired, the EI values will rise while the EV values will decline.

From a clinical perspective, it may be beneficial to consider the complementary use of both the EV and EI in assessing muscle tissue characteristics. While we have presented evidence supporting the relevance of the EV, the EI remains a familiar metric among physiotherapists. Utilizing both may mitigate the limitations inherent to each when assessed in isolation, thereby reducing potential biases and enhancing the overall characterization of muscle tissue. In light of this, the EV shows great potential for practical applications in clinical settings, particularly as an indicator for monitoring rehabilitation progress and guiding therapeutic decisions. By providing quantitative assessments of muscle tissue quality, the EV complements existing qualitative scales, enabling the detection of subtle changes in muscle composition. For instance, during the rehabilitation process, changes in the EV values may help a clinician quantify the treatment’s effects over time or the adverse effects of some treatments, such as botulinum toxin injection. Furthermore, the EV could serve as a diagnostic tool for stratifying patients according to the severity of muscle impairment, helping to develop personalized rehabilitation plans and more targeted therapeutic strategies [43]. Our findings establish EV as a promising indicator, but its clinical implementation requires further validation and research.

Our results are similar to those observed in individuals with amyotrophic lateral sclerosis, where the muscles affected also show an increased EI and a reduced EV [44]. However, in contrast to our findings, GLCM features have been identified as relevant in amyotrophic lateral sclerosis [14]. Similarly, the EV has been shown to discriminate between different types of soleus injuries based on the EI patterns [45]. It has also been used to differentiate between the plantar extrinsic musculature and the plantar fascia in individuals with plantar fasciitis compared to healthy controls [28,46], as well as in individuals with patellar tendinopathy who have a higher EV compared to healthy controls [47].

One of the challenges of a texture analysis is knowing how to handle the large amount of data that can be extracted from a single image, as there can be hundreds of parameters, which means that, even for that reason alone, high correlations can be found between them [4]. In our study, we reduced twenty-one initial features to just one, combining an exploratory analysis and different factors that may affect the possibility of comparing the results and translating them into clinical practice. Another critical challenge is the significant variability between the programs/software that calculate these echotexture features, which makes it difficult to compare and interpret the results. However, concerning the first-order features, an excellent concordance has been demonstrated between software packages [48]. This reinforces our results regarding considering the EV (first-order feature) as the main feature for characterizing muscle tissue quality in individuals who have suffered a stroke.

The results presented in this study mix a clinical point of view, based on expert physiotherapists, with a technical point of view, based on a data analysis and programming without any data pre-processing. The main strengths of our study are the absence of possible biases in the selection of the ROI, with the collection of the maximum amount of muscle tissue to prevent the loss of relevant information, together with the standardization of the US image collection protocol and the clarity with which all the settings, features, and programs used have been shown. However, the results of this study are subject to certain limitations inherent in its observational design. Firstly, various factors that could affect US imaging, such as the age [49], sex [6], muscle strength [50], amount of skin, fascia, subcutaneous adipose tissue [51,52], genetic and epigenetic components [53], time from stroke onset, and prior therapeutic interventions received, among others, were not considered. We considered the standardized image collection procedure to be a strength, although it could also be understood as a limitation, since we do not know if it can be extrapolated to the rest of the gastrocnemius muscle areas or if there are differences between both sides or between different muscles. However, considering that this is a chronic condition, we assumed that the degree of impairment would be similarly distributed throughout the muscle, which does not occur in other conditions like muscle tears. Furthermore, despite the large number of US images used, they were only extracted from 22 stroke-affected individuals, which is a notable limitation of the study, as it restricts the generalizability of the findings to the general population. However, the results are a starting point for other studies on the same population and on populations with different pathologies. Lastly, another limitation is the absence of formal inter- and intra-rater reliability testing for the muscle tissue quality assessments. Although minimal inter-rater discrepancies were observed, conducting such assessments in future research would strengthen the robustness and reproducibility of the findings.

Although this was an observational study in which the capacity of different echotexture features to characterize the gastrocnemius medialis muscle tissue quality in stroke-affected individuals was studied, it could be interesting to explore their use in the follow-up and evolution of patients and the analysis of the effectiveness of treatments, as has been proposed for other echotexture features in different populations [54].

## 5. Conclusions

Based on a mixed technical and clinical approach, the EV proved to be the most informative echotexture feature for quantifying the degree of muscle tissue impairment in the gastrocnemius medialis of post-stroke individuals, with a higher EV indicating a better muscle tissue quality. The EV represents a cost-effective and non-invasive method for assessing muscle tissue characteristics, providing a valuable alternative to other imaging modalities. Furthermore, the EV offers a promising tool for monitoring stroke rehabilitation and could be explored in other neuromuscular conditions.

## Figures and Tables

**Figure 1 jcm-13-07800-f001:**
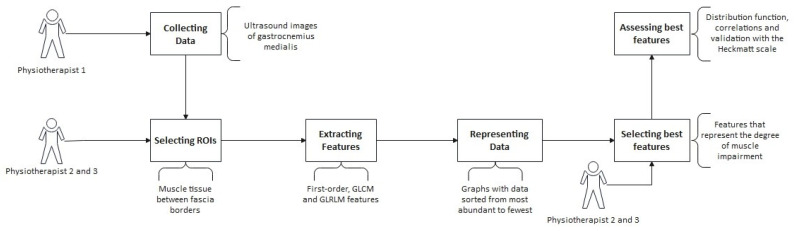
Research pipeline.

**Figure 2 jcm-13-07800-f002:**
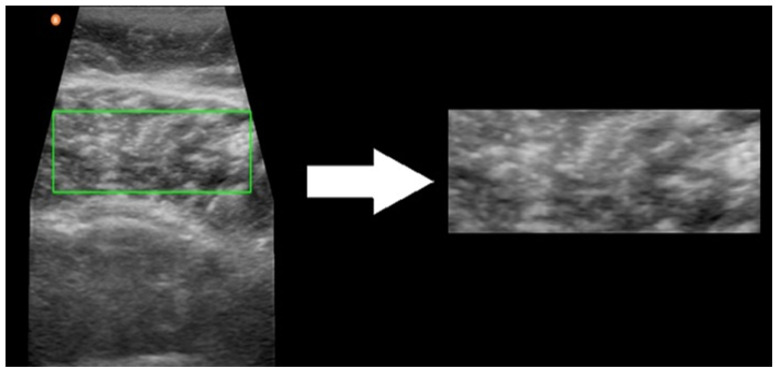
Manual ROI selection process with as much muscle tissue as possible. The green rectangle indicates the final selection of the ROI after consensus between two physiotherapists.

**Figure 3 jcm-13-07800-f003:**
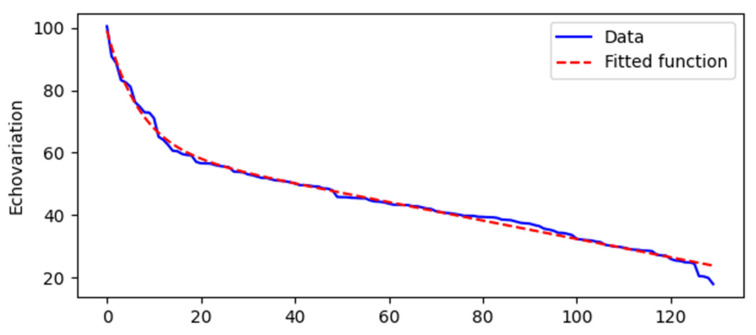
Graph showing the fitted distribution of EV values across ROIs. The scale on the X-axis shows the number of the US image, while the Y-axis shows the echovariation (EV) values.

**Figure 4 jcm-13-07800-f004:**
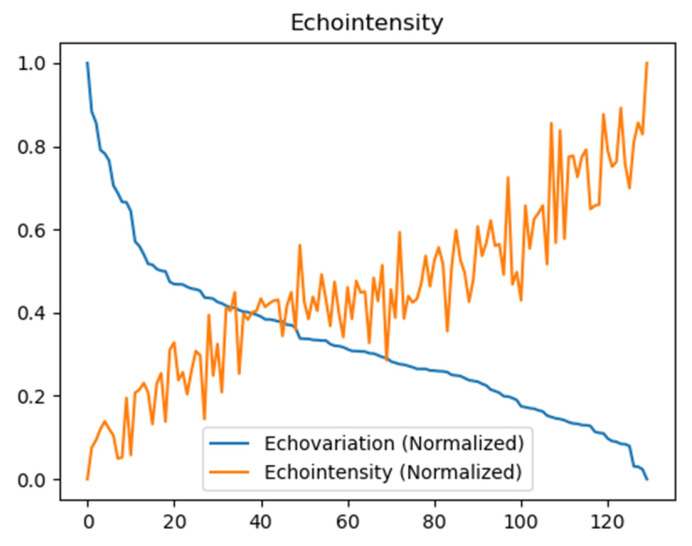
Graphical representation of the correlation between echovariation (EV) and echointensity (EI), the only one that showed a Pearson’s correlation coefficient > 0.80. The scale on the X-axis shows the number of the US image, while the Y-axis shows the echovariation (EV) and echointensity (EI) values normalized on a scale from 0 to 1.

**Figure 5 jcm-13-07800-f005:**
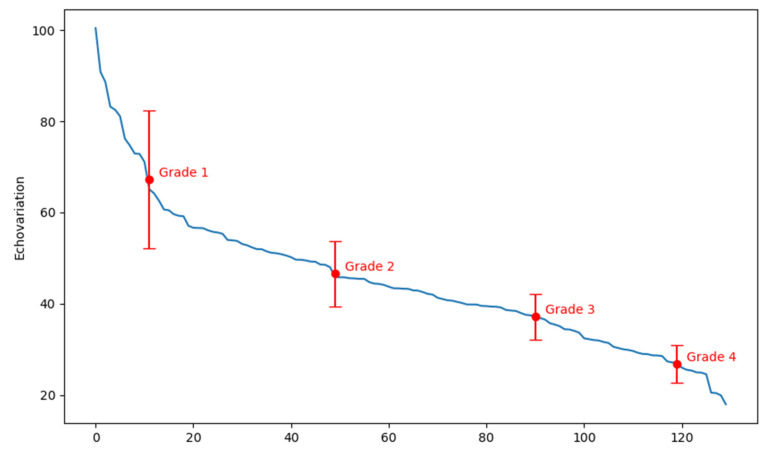
Distribution of the four grades of the modified Heckmatt scale along echovariation (EV) values, with mean and standard error bars for each grade. The scale on the X-axis shows the number of the US image, while the Y-axis shows the echovariation (EV) values.

**Table 1 jcm-13-07800-t001:** Description of the echotexture features extracted from each of the US images.

Statistical Approach	Feature	Brief Description
Histogram-based (first-order) features	Echogenicity, also called echointensity (EI)	The ability of tissues to reflect ultrasound waves. Specifies the average brightness of the ultrasound image.
	Variance	Measures the dispersion of pixel intensity values.
	Standard deviation	The square root of variance.
	Echovariation (EV), also called coefficient of variation	Variation in echointensities in an ultrasound image.
	Skew	Measures the asymmetry of the distribution of pixel intensity values.
	Kurtosis	Measures the “tailedness” of the distribution of pixel intensity values.
Grey-level co-occurrence matrix (GLCM)	Correlation	Measures the statistical correlation between neighboring pixels.
	Dissimilarity	Measures the mean absolute difference between neighboring pixel values.
	Contrast	Measures the intensity contrast between a pixel and its neighbor across the ROI.
	Homogeneity, also called inverse different moment	Measures the closeness of the distribution of elements in the GLCM to the GLCM diagonal.
	Angular second moment (ASM)	Similar to energy, it measures the uniformity of the GLCM.
	Energy	Sum of squared elements in the GLCM, representing texture uniformity.
	Max probability	The highest value in the GLCM.
	Entropy	Measures the randomness or complexity of the texture.
	Cluster shade	Measures the skewness of the distribution of the GLCM.
	Cluster prominence	Measures the peakedness of the distribution of the GLCM.
Grey-level run length matrix (GLRLM)	Short run emphasis (SRE)	Measures the distribution of short runs.
	Long run emphasis (LRE)	Measures the distribution of long runs.
	Grey-level uniformity (GLU)	Measures the uniformity of grey levels.
	Run length uniformity (RLU)	Measures the uniformity of run lengths.
	Run percentage (RPC)	The ratio of the number of runs to the number of pixels.

**Table 2 jcm-13-07800-t002:** The evaluation process was conducted by two physiotherapists using ultrasound images with the top and bottom 10 values of each echotexture feature. The number of images was categorized based on consistency with more affected muscle tissue patterns.

Features	Top 10 Values		Bottom 10 Values	
	Physiotherapist 1	Physiotherapist 2	Physiotherapist 1	Physiotherapist 2
Echointensity *	10/10	10/10	0/10	0/10
Variance	8/10	10/10	7/10	8/10
Standard deviation	9/10	10/10	7/10	7/10
Echovariation *	0/10	0/10	10/10	10/10
Skew	2/10	3/10	10/10	10/10
Kurtosis *	0/10	0/10	10/10	10/10
Correlation	8/10	8/10	4/10	3/10
Dissimilarity	1/10	1/10	5/10	5/10
Contrast	3/10	4/10	5/10	5/10
Homogeneity	2/10	2/10	5/10	5/10
ASM	3/10	4/10	7/10	9/10
Energy	0/10	0/10	5/10	7/10
Max probability	0/10	0/10	7/10	8/10
Entropy	7/10	7/10	8/10	8/10
Cluster shade	8/10	10/10	2/10	2/10
Cluster prominence	7/10	10/10	1/10	1/10
SRE	8/10	9/10	7/10	8/10
LRE	3/10	3/10	7/10	9/10
GLU	8/10	8/10	10/10	10/10
RLU	10/10	10/10	10/10	10/10
RPC	10/10	10/10	9/10	9/10

Features marked with an asterisk (*) indicate those that met the criteria to be selected as the most informative features. For example, an indicator showing a score of 0/10 for the top 10 values and 10/10 for the bottom 10 values (or vice versa) would be optimal.

**Table 3 jcm-13-07800-t003:** Correlations between each echotexture feature with echovariation (EV) as a reference.

Feature	Correlation	*p*-Value
Echointensity	−0.90 *	<0.001 ^a^
Variance	0.34	<0.001 ^a^
Standard deviation	0.35	<0.001 ^a^
Skew	0.68	<0.001 ^a^
Kurtosis	0.46	<0.001 ^a^
Correlation	0.21	0.019 ^a^
Dissimilarity	0.03	0.753
Contrast	0.18	0.040 ^a^
Homogeneity	0.53	<0.001 ^a^
ASM	0.62	<0.001 ^a^
Energy	0.65	<0.001 ^a^
Max probability	0.72	<0.001 ^a^
Entropy	−0.20	0.024 ^a^
Cluster shade	−0.23	0.009 ^a^
Cluster prominence	−0.37	<0.001 ^a^
SRE	−0.12	0.190
LRE	0.38	<0.001 ^a^
GLU	−0.14	0.116
RLU	−0.08	0.349
RPC	0.03	0.740

^a^ Statistically significant differences (*p*-value < 0.05); * correlations with a Pearson’s coefficient > 0.80.

**Table 4 jcm-13-07800-t004:** Descriptive data and comparison analyses of echovariation (EV) values across the four grades of the modified Heckmatt scale.

Feature	Grade 1 (*n* = 24)	Grade 2 (*n* = 58)	Grade 3 (*n* = 24)	Grade 4 (*n* = 24)	Between-Group Comparison (Kruskal–Wallis Test)	Post Hoc Analysis (Wilcoxon Tests with Bonferroni Correction)
Echovariation	67.28 ± 15.05	46.62 ± 7.17	37.16 ± 4.94	26.85 ± 4.06	<0.001 ^a^	Grade 1 vs. Grade 2: <0.001 ^a^ Grade 1 vs. Grade 3: <0.001 ^a^ Grade 1 vs. Grade 4: <0.001 ^a^ Grade 2 vs. Grade 3: <0.001 ^a^ Grade 2 vs. Grade 4: <0.001 ^a^ Grade 3 vs. Grade 4: <0.001 ^a^

Data are presented as mean ± standard deviation; ^a^ statistically significant difference (*p*-value < 0.05).

## Data Availability

The data presented in this study are available from the corresponding author upon request, due to the fact that these data are part of a larger, ongoing project and may not yet be fully available for public sharing until the project is completed.

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
