# Peer review of "Characterizing Muscle Tissue Quality Post-Stroke: Echovariation as a Clinical Indicator"

_jcm, 2024, doi:10.3390/jcm13247800_

Round 1
Reviewer 1 Report
Comments and Suggestions for Authors
Q1: abstract: Results could be more precise. For instance, instead of "EV shows promise," state "EV demonstrated significant correlation with the Heckmatt scale (r = -0.81, p < 0.001)."
Q2: Introduction:: Line 87: “EV, which is statistically defined by the ratio of the standard deviation to the mean pixel intensity,” is wordy. Simplify for better readability.
Q3: Redundancies remove. Remove less relevant details, focusing on US imaging and EV.
Q4. Methods: Lack of clarity about ROI (Region of Interest) selection criteria beyond consensus by physiotherapists.
Q5: Were quantitative thresholds applied to standardize the ROI size?
Q6: Methods: Were the physiotherapists blinded to clinical outcomes during classification?
Q7: Methods: Could inter-observer variability during ROI selection be quantified statistically?
Q8: Methods: Mention explicitly why certain statistical tests (e.g., Kruskal-Wallis) were chosen over others.
Q9. Results Table 2: The "Highest" and "Lowest" value columns are difficult to interpret. Clarify how physiotherapists determined categorization consistency.
Q10:Line 279: "mathematically encompasses EI" – rephrase to “includes EI in its calculation.”
Q11: Line 152-153: “with any image pre-processing” should likely be “without any image pre-processing.”
Q12: Line 179: "each feature’s 10 highest and 10 lowest values" – rephrase to avoid awkward phrasing (e.g., “the top and bottom 10 values of each feature”).
Q13: Figures and Tables Add a new table or figure summarizing key EV-related findings across modified Heckmatt scale grades.
Q14: Figures and Tables Supplementary material: Explicitly link Figures S1 and S2 to findings in Figure 3.
Q15: Line 303: "Figure 3. Fitted function on the distribution..." could be rephrased to, “Graph showing the fitted distribution of EV values across ROIs.”
Q16: Discussion: The "S, L" hypothesis (Lines 385–389) is too technical for most clinical audiences.
Q17: Line 373: “confirms the use of this biomarker” should be “supports the use of this biomarker.”
Q18: Consider adding practical applications for EV in clinical settings, such as its use in rehabilitation progress tracking or as a diagnostic aid.
Q19. Conclusion : Consider emphasizing the significance of the findings: “EV offers a cost-effective and non-invasive method for muscle quality assessment.”
Q20. Conclusion : could emphasize the potential translational impact of EV in clinical settings: “EV offers a promising tool for monitoring stroke rehabilitation and could be explored in other neuromuscular conditions.”
Q21: limitation:Small sample size (22 participants) limits the generalizability of findings. Acknowledge this limitation more prominently.
Q22: limitation: The absence of inter- and intra-rater reliability testing is a potential oversight.
Comments on the Quality of English LanguageThe English could be improved to more clearly express the research.
Author Response
We attach a reply for both reviewer 1 and 2

Reviewer 2 Report
Comments and Suggestions for Authors
First of all I want to thank the authors for giving me the opportunity to review their article.
The article is interesting, but after reading it I have some concerns and some questions that I will detail below:
Is this US marker really a BIOmarker? I would keep just marker.
The abstract gives important information. I would not change anything in it.
Introduction: I think the introduction is too large and it gives too many details about subjects that are not subject of this article.
Line 48-61 - I would try to put the information in this paragraph in just a phrase. It gives too many information that is not quite the subject of the article.
Line 72-98 - I think it would be better if the paragraphs would be shorter. They have too much information and the readers will pass this part without giving too much attention.
Materials and Methods
In order to understand the study a figure that describes the study outline should be included: total number of patients included in the study, inclusion and exclusion criteria (which can be afterwards included in a table), the causes of exclusion of the patients, etc.
I do not understand how were the patients included in the study. Were the patients chosen or all patients admitted in the hospital (that met the inclusion criteria) were included in the study?
Table 1 is too exhaustive and Formula does not bring any special clarifications.
The methodology should be rethought: it is not clear and it is too large, with too much information. Maybe it will help here a figure too. It would include all information in a more clear way.
Why were included in the study only patients that can walk? I think this information should be discussed.
It is not provided in the methodology information about the onset of the stroke (I understand that 6 months after the onset, but I think it is important for the 22 participants how many were at 6 months and how man were 3 years after the stroke for example).
It is important to specify the type of stroke (hemorrhagic or ischemic).
It is not specified in the article what medication received the participants (it is important to know if there were participants receiving baclofen for example).
Results
Should be added comorbidities of the patients included in the study.
Line 276-287: should be moved to discussions.
Discussions
Should be added more similar or opponent studies to prove the need for such a marker. Discussions should be a text with pro and cons (Did other authors approach same marker? Did they have similar results?) and not just another description of the study itself.
Conclusions
Conclusions are clear and they summarize the entire article.
I think the references should be mentioned at the end of the phrase.
Author Response

(The authors gave the same response as above.)

Round 2
Reviewer 1 Report
Comments and Suggestions for Authors
accepted
Reviewer 2 Report
Comments and Suggestions for Authors
Dear authors,
Thank you for responding to all of my concerns about your manuscript.